anxiety; personal resources; emotional exhaustion; sleep quality; internal locus of control; emotional intelligence

**Corresponding author:**
Sabine Saade;
Email: ss241@aub.edu.lb

# Exploring the links between workers' personal resources and emotional exhaustion during challenging times: Anxiety as a mediating factor

Annick Parent-Lamarche[1] and Sabine Saade[2] 

[1]Université du Québec à Trois-Rivières, Canada and [2]American University of Beirut, Lebanon

## Abstract

**Purpose:** Using a cross-sectional design, this study aimed to examine the associations between personal resources and emotional exhaustion, with anxiety as a potential variable consistent with a mediating role.

**Methods:** Data was collected in Lebanon over a six month period using validated self-report questionnaires. Workers aged 18 to 64 years ($N = 295$) were recruited using a non-randomized snowball sampling approach. Multiple regression and mediation analyses were conducted.

**Results:** The findings indicate that personal resources (sleep quality ($b = -0.224$, 95% CI $[-0.286, -0.165]$), emotional intelligence ($b = -0.061$, 95% CI $[-0.112, -0.007]$), and internal locus of control ($b = -0.216$, 95% CI $[-0.351, -0.075]$) were all negatively associated with anxiety, supporting Hypothesis 1. Sleep quality ($b = 0.073$, 95% CI $[-0.125, -0.029]$) and internal locus of control ($b = -0.071$, 95% CI $[-0.140, -0.018]$)) were also associated with lower emotional exhaustion through their associations with lower anxiety levels (i.e., indirect association via anxiety). In contrast, emotional intelligence ($b = -0.020$, 95% CI $[-0.046, 0.002]$) showed no significant indirect association with emotional exhaustion (i.e., no indirect association via anxiety).

**Conclusion:** The results of this study highlight that not all personal resources have uniformly positive effects.

## Impact statements

This study highlights the critical role of personal resources in supporting psychological well-being in high-stress work environments, using Lebanon as a context of chronic socio-economic and political adversity. By examining factors such as sleep quality, internal locus of control, and Emotional Intelligence (EI), this study demonstrates how these resources are associated with lower anxiety and reduced EI, under challenging circumstances. Importantly, the findings obtained reveal nuanced effects regarding EI. While "regulation of emotion" supports psychological health, strategic or purely instrumental "use of emotion" may have unintended negative effects, emphasizing the need for carefully designed interventions. This finding underscores that promoting personal resources is not only about increasing skills but also about encouraging their adaptive application in everyday work life. From a practical standpoint, organizations and policymakers can draw on these findings to implement programs that could help enhance resilience and promote psychological health. Initiatives could include promoting sleep hygiene awareness, holding workshops to foster internal locus of control, and offering EI trainings. In a context of chronic adversity, these strategies can help employees better manage stressors, improve workplace performance, and maintain psychological health. Beyond Lebanon, this study contributes to a better understanding of how personal resources operate under chronic stress, providing guidance for workplaces facing similar environmental or societal pressures. By clarifying which resources play a protective role and how they can be cultivated, this study informs both research and practice, particularly when facing a complex organizational and societal context. Lastly, the results obtained could help advance strategies to enhance workers' psychological health.

## Introduction

Daily life is fraught with obstacles and challenges that can undermine an individuals' psychological health, including that of workers. That is particularly true during periods of global turmoil, as witnessed in recent years. Notably, the COVID-19 pandemic and, more recently, a severe economic crisis have had a profound impact on the Lebanese population (Hassan, 2022). In the face of such disruptive events, individuals must demonstrate resilience – or at the very least, make efforts to cultivate it – in order to cope with those stressors (Friedberg and Malefakis, 2022).

Stressful events tend to increase an individuals' vulnerability to "loss spirals," exacerbating risks to their psychological health (Hobfoll et al., 1991). A study conducted during the pandemic found that locus of control (an important personal resource) significantly shifted from internal to external among both students and professionals (Misamer et al., 2021). Stressful events of this magnitude are likely to elicit a sense of powerlessness and a loss of control, which can further compromise psychological health (Misamer et al., 2021). As for Lebanon, 77.78% of individuals reported experiencing work-related burnout (El Hachem and Atallah, 2022). The country has long borne a disproportionately high burden of anxiety, with a lifetime prevalence rate of 25.8% (Karam et al., 2008), primarily driven by cumulative adversity and recurrent national crises (Farran, 2021). Amid limited access to personal funds and a deteriorating economic situation, many Lebanese workers were compelled to return to the workforce after retirement or to on take on additional jobs (Yacoub et al., 2023). Recent studies indicate that workers in Lebanon are experiencing higher workloads (Islam et al., 2021), increased stress (Yacoub et al., 2023), greater fatigue (Shallal et al., 2021), as well as job insecurity and salary reductions (Islam et al., 2021), all of which contribute to their vulnerability to burnout and anxiety. This context underscores the urgent need to identify and strengthen protective factors that can shield individuals from anxiety and burnout. This is especially true in times of heightened adversity, when coping becomes significantly more challenging. In this context, the scientific literature emphasizes the crucial role of personal resources in promoting psychological health and preventing conditions such as burnout.

Burnout is a work-related condition marked by prolonged mental and emotional exhaustion resulting from sustained coping efforts in response to job stressors (Maslach et al., 2001). This condition typically comprises three dimensions: emotional exhaustion, cynicism, and a diminished sense of professional efficacy, with emotional exhaustion being its most central and visible manifestation (Maslach et al., 2001). In the present study, we focused specifically on emotional exhaustion, as it represents the core component of burnout, and the most direct indicator of strain.

One key determinant of psychological health is sleep quality. Defined as an individual's overall satisfaction with their sleep experience, including ease of falling and staying asleep, total sleep duration, and feeling refreshed upon waking (Nelson et al., 2022), is widely recognized as a key component of psychological health, particularly due to its role in emotional regulation, cognitive functioning, and recovery from daily stressors (Mauss et al., 2013; O'Leary et al., 2017). Alongside diet and exercise, sleep is considered one of the three essential pillars of health (Shechter et al., 2014). Research has consistently shown that sleep quality is associated with both physical and psychological health-related qualities of life among young adults (Clement-Carbonell et al., 2021). Importantly, the relationship between sleep and psychological health seems to be bidirectional: disturbances in sleep are closely linked to symptoms of anxiety and depression, and vice versa (Alvaro et al., 2013). Poor sleep quality, for example, often arises in the context of stressful experiences. Studies have found that exposure to social stressors at work can predict poorer sleep quality (Pereira et al., 2016), and in turn, poor sleep can be associated with heightened levels of negative affect the following day (Marcusson-Clavertz et al., 2022). Similarly, individuals deprived of adequate sleep tend to display intensified negative emotional responses to mild stressors (Minkel et al., 2012). Stressors not only worsen sleep quality but can also disrupt the circadian and homeostatic systems that regulate sleep (Han et al.,

2012). In cases of stress-related poor sleep quality, a vicious cycle can develop through the activation of the hypothalamic–pituitary–adrenal axis, further compromising sleep and psychological health (Han et al., 2012). As such, good sleep quality can be considered a personal resource.

EI is another key personal resource, encompassing abilities that enable individuals to accurately recognize and understand emotions, both in themselves and in others. EI allows us to regulate our emotional responses effectively and use emotions to facilitate motivation, planning, and goal-directed behavior (Salovey and Mayer, 1990). EI encompasses four dimensions: self-emotion appraisal, others' emotion appraisal, use of emotion, and regulation of emotion. Self-emotion appraisal refers to the ability to identify and understand one's own emotional states, supported by introspection and the formation of coherent interpretations based on emotional self-awareness (Davies et al., 1998; Law et al., 2004). Individuals' adept in this area tend to express emotions clearly, facilitating healthy interpersonal interactions. Others' emotion appraisal involves perceiving and understanding the emotions of those around us, an ability often described as empathy, which aids in predicting emotional responses and selecting socially appropriate behaviors (Davies et al., 1998; Law et al., 2004). The use of emotion refers to channeling emotional experiences into constructive action, such as sustaining motivation and resilience in the face of challenges (Davies et al., 1998; Law et al., 2004). Finally, regulation of emotion involves effectively managing emotional reactions to promote quicker recovery from distress. It also includes the ability to modulate emotional expression to shape others' impressions, thereby fostering more adaptive and positive mood states (Davies et al., 1998; Law et al., 2004). High levels of EI have been found to be associated with positive psychological health outcomes in various populations, including students (Di Fabio and Kenny, 2016; Moeller et al., 2020), adolescents (Guerra-Bustamante et al., 2019), and workers (Ordu et al., 2022). This resource enables individuals to interpret stressors in an adaptive manner and maintain a sense of personal agency.

As mentioned earlier, locus of control, which reflects how individuals perceive their ability to influence life events, is also one such personal resource. People with an internal locus of control generally believe that their actions, decisions, and abilities primarily shape their major life outcomes, rather than external factors such as luck or fate (Rotter, 1966). Like EI, this internal orientation has been consistently linked to better mental health outcomes in both the general adult population (Sigurvinsdottir et al., 2020; Shin and Lee, 2021) and among workers (Parent-Lamarche and Marchand, 2019).

Another key factor influencing psychological health is anxiety, which refers to a persistent state of apprehension and uncontrollable worry that interferes with daily functioning (Stein and Sareen, 2015). Anxiety can gradually erode personal resources, making individuals more vulnerable to burnout over time (Hobfoll and Shirom, 2000; Childs et al., 2024).

Despite growing recognition of the importance of personal resources, their protective role remains insufficiently explored in contexts of extreme adversity, such as the one currently faced by workers in Lebanon. Understanding how these resources operate in such circumstances is particularly important, especially given the rise in major global stressors in recent years. Beyond the pandemic, many populations are contending with the repercussions of armed conflicts, severe economic crises, and political instability. For instance, growing uncertainty stemming from shifts in U.S. leadership has triggered trade tensions and market volatility (Bown, 2025;

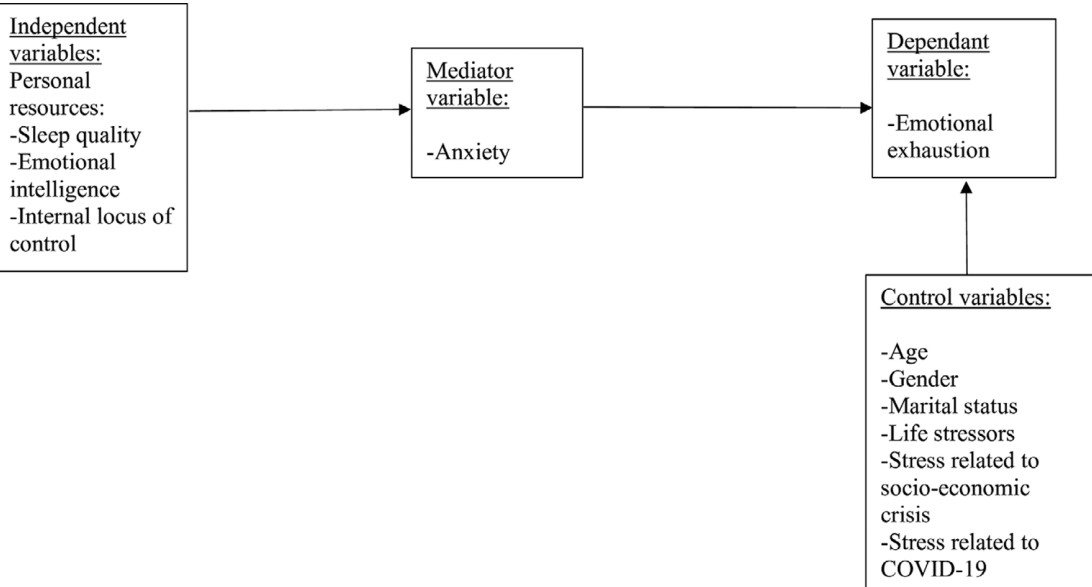

**Figure 1.** Conceptual model

International Monetary Fund, 2025; Statista Research Department, 2025), generating ripple effects that may further destabilize fragile economies worldwide, including Lebanon. Those stress-inducing circumstances have not only become more frequent but also more intense, compounding the already precarious conditions faced by Lebanese workers.

In line with the Conservation of Resources (COR) theory (Hobfoll, 1989), we propose that in high-stress environments, personal resources such as sleep quality, EI, and internal locus of control help reduce anxiety, thereby reducing emotional exhaustion.

Building on this theoretical model, the objective of the present study was to examine how personal resources, specifically sleep quality, EI, and locus of control influence anxiety. We were also interested in examining whether anxiety shows an indirect association with the link between these resources and emotional exhaustion.

As such, this study addresses the following research questions: how do sleep quality, EI, and locus of control influence anxiety? Does anxiety is *consistent* with a mediating role in the relationships between personal resources (i.e., sleep quality, emotional intelligence, and locus of control) and emotional exhaustion?

By investigating these associations, we aim to inform prevention and intervention strategies. Strategies that enable individuals and organizations to better manage stressors, address anxiety-related challenges, and foster the development of key personal resources. Please see Figure 1.

### Theoretical background and hypotheses development

The theoretical framework underpinning our model is grounded in the COR theory (Hobfoll, 1989). As Van Woerkom et al. (2016) emphasize, COR theory provides a nuanced understanding of how individuals manage personal resources in demanding environments. Our central proposition is that, in the face of persistent

psychological stressors, the presence or absence of personal resources – specifically sleep quality, EI, and internal locus of control – significantly influences psychological health outcomes. Those resources are crucial for helping individuals manage stress, build resilience, and maintain psychological health. Sleep quality restores the physical, emotional, and cognitive capacities needed to cope with daily stressors (Mauss et al., 2013; O'Leary et al., 2017), while EI enables individuals to recognize and understand emotions, regulate EI responses, and leverage emotions to enhance motivation and goal-directed behavior. In high-stress contexts like the COVID-19 pandemic and Lebanon's socio-economic crisis, these resources are particularly important for mitigating negative psychological effects (Kniffin et al., 2021). As mentioned earlier, locus of control reflects an individual's perception of their ability to influence life events (Rotter, 1966). Individuals with an internal locus of control believe that their actions and decisions primarily shape their outcomes, rather than relying on external factors like luck or fate. This resource is therefore particularly valuable in navigating adversity. In line with the COR theory, these personal resources serve as upstream protective factors that reduce the likelihood of entering a cycle of psychological strain. Individuals with strong resources are better equipped to regulate their emotional responses to stressors. Conversely, limited resources increase vulnerability to anxiety. While other psychological states such as depression may similarly function as mediators, anxiety typically occurs earlier in response to resource depletion (Starr and Davila, 2012), whereas stress can be conceptualized more as a reaction to specific sources or demands rather than a consequence of resource loss (Jex et al., 1992). In line with the COR theory, vulnerability to anxiety functions as an early indicator of resource depletion and initiates a loss spiral, where initial psychological strain leads to further erosion of resources and ultimately to emotional exhaustion. In our model, anxiety is conceptualized as a variable consistent with a mediating role between personal resources and emotional exhaustion. In our model, anxiety is conceptualized as a variable playing a mediating role between personal resources and emotional exhaustion. In the first phase of this sequence, anxiety is considered a variable consistent with a dependent variable,

allowing us to examine the associations between independent variables and this proposed mediator. In turn, this mediator serves as a variable consistent with a mediating role in the full model linking personal resources to emotional exhaustion. Regarding testing H1, anxiety was treated as the dependent variable. This is common in sequential effect analyses where direct effects are first assessed on the mediator (i.e., anxiety, treated as the DV in the first sequence of regressions, which does not include emotional exhaustion) before examining indirect effects on the ultimate outcome (emotional exhaustion) in the second sequence of regressions (Preacher and Hayes, 2004). This approach is particularly informative. More specifically, this method allows us to isolate the direct influence of personal resources on anxiety before evaluating how anxiety subsequently transmits these effects to emotional exhaustion. This model provides us with a clearer understanding of the mechanisms underlying psychological strain. As individuals experience heightened anxiety due to weaker personal resources, they expend greater psychological energy attempting to cope, intensifying their risk of suffering from emotional exhaustion. From a COR perspective, this reflects a dynamic of escalating loss: initial resource depletion (e.g., poor sleep, diminished emotional regulation, externalized locus of control) leads to anxiety (H1), which in turn accelerates emotional exhaustion (H2). Based on these theoretical foundations, we propose that:

**H1 (Direct Association):** Personal resources are directly and negatively associated with anxiety.

**H2 (Consistent with a Mediation Effect):** Personal resources are indirectly and negatively associated with emotional exhaustion via their effect on anxiety.

## Method

### Participants and procedure

Data collection was carried out over a six-month period, spanning from February to August 2022, as part of a cross-sectional study. We recruited 295 workers from various sectors residing across Lebanon and employed at the time of the study. The main inclusion criteria included being literate in either English or Arabic, aged between 18 and 64 years, and currently employed at the time of participation. Participants who did not meet these criteria, for example, individuals younger than 18, older than 64, or not currently employed, were excluded from the study. To reach a diverse sample, we used a non-randomized snowball sampling approach. Initial participants from various professional roles and employment sectors were invited to refer colleagues and peers. This helped us broaden the sample across different regions and occupations. This approach was particularly suitable given the challenges of accessing a comprehensive list of all workers in Lebanon. Overall, missing data were generally low to moderate, including anxiety, emotional exhaustion, sleep quality, internal locus of control, and EI. Missing data patterns were first examined in Stata using misstable patterns, which showed that the majority of cases were complete and that missingness was dispersed across multiple patterns. Little's Missing Completely At Random (MCAR) test conducted on the key study variables ($\chi^2$ = 98.51, df = 93, $p$ = 0.3281) indicated that the data can be considered MCAR. Based on these results and the relatively modest levels of missingness, listwise deletion was applied during preprocessing in Stata, and the resulting dataset without missing values was

subsequently imported into Mplus for analysis. To ensure inclusivity and minimize barriers related to language proficiency, access to electricity, or Internet availability, participants were offered the flexibility to complete the survey either online or via paper format, in the language of their choice (English or Arabic). To ensure accessibility for individuals from diverse socio-economic backgrounds across Lebanon, all study materials were translated into Arabic. We employed the standard translation and back-translation procedure recommended by Bracken and Barona (1991). Initially, a student specializing in Arabic at the American University of Beirut translated the materials from English to Arabic. A second Arabic major, unfamiliar with the original content, then independently back-translated the materials into English. Any inconsistencies identified between the original and back-translated versions were reviewed and resolved collaboratively to ensure accuracy and conceptual equivalence.

Several measures were implemented to protect participant confidentiality. Individuals completing paper versions of the survey were provided with private space and sufficient time to answer freely. As an incentive, participants were entered into a lottery for a chance to win one of two cash prizes of 400,000 Lebanese pounds. Prior to participating, members of the research team reviewed the consent form with each participant and clearly outlined the measures taken to protect confidentiality. For individuals completing the study online, explicit confirmation of having read and understood the consent form was required before proceeding with the study. Once informed consent was obtained, participants were invited to complete a series of questionnaires. On average, the entire process took approximately 30 min to complete.

The sample was notably diverse, encompassing individuals employed across a wide spectrum of occupations and industries. Participants represented 19 job sectors, including accommodation and food services, administrative and support services, agriculture, arts and entertainment, construction, finance and insurance, healthcare and social assistance, information and cultural industries, manufacturing, professional and technical services, public administration, education, retail, real estate, and transportation. The most represented sectors were teaching services (12.54%); health care and social assistance (11.53%); professional, scientific, and technical services (7.12%); public services (5.76%); other services excluding public administration (5.08%); administrative and support services (4.07%); management of companies and enterprises (4.07%); arts, entertainment, and recreation (4.75%); finance and insurance (3.39%); and public administration (3.05%).

Participants held a broad range of roles, reflecting a diverse occupational mix, including University professors, librarians, lawyers, HR directors, photographers, hairdressers, curators, creative directors, business analysts, baristas, receptionists, behavior analysts, auditors, architects, bankers, and administrative assistants, among others.

### Measures

#### Emotional exhaustion

Emotional exhaustion, which served as the dependent variable in our study, was measured based on a five-item (e.g., "Working all day is really a strain for me"; α = 0.95) seven-point additive scale from Schaufeli (1996). Possible answers ranged from 0 (never) to 6 (daily). It was treated as a continuous variable, with possible total scores ranging from 0 to 30, with higher scores indicating greater emotional exhaustion. To the best of our knowledge, no published Arabic validation of the MBI-GS exists. Accordingly, we used a

translated and back-translated version following the method proposed by Bracken and Barona (1991) to ensure linguistic and conceptual equivalence.

### Anxiety

Anxiety, which served as a variable consistent with a mediator in this study, was measured based on a seven-item (e.g., "Over the last 2 weeks, how often have you been bothered by the following problems: Worrying too much about different things"; α = 0.94) four-point additive scale from Spitzer et al. (2006). Possible answers ranged from 0 (not at all) to 3 (nearly every day). It was treated as a continuous variable, with possible total scores ranging from 0 to 21, with higher scores indicating greater anxiety. Note that the Arabic version has also been validated, showing adequate reliability and validity for use in Arabic-speaking populations (Khoury-Malhame et al., 2025).

### Sleep quality

Sleep quality, which served as one of the independent variables in this study, was measured based on a six-item (e.g., "During the past month, how many days a week has it taken you more than 30 min to fall asleep after the light was switched off?"; α = 0.85) seven-point additive scale from Pallesen et al. (2008). Possible answers ranged from 0 to 7. The Bergen Insomnia Scale (BIS) (Pallesen et al., 2008) is typically used to measure insomnia symptoms, with higher scores indicating poorer sleep. In our study, we reverse-coded the BIS items so that higher scores represent better sleep quality, aligning the measure with our conceptualization. Cronbach's alpha remained unchanged after reversal (α = 0.85), indicating stable internal consistency. The scale was treated as a continuous variable, with total scores ranging from 0 to 42, representing the full spectrum from poorest to best sleep quality in our sample. To the best of our knowledge, no published Arabic validation of the BIS exists. Accordingly, we used a translated and back-translated version following the method proposed by Bracken and Barona (1991) to ensure linguistic and conceptual equivalence.

### Emotional intelligence

EI, which served as one of the independent variables in our study, was evaluated based on the Wong and Law (2002) EI scale, which includes 16 items (e.g., "I am sensitive to the feelings and emotions of others." α = .90) using a seven-point additive scale ranging from 1 (do not at all agree) to 7 (very strongly agree). This variable was treated as a continuous variable, with possible total scores ranging from 16 to 112, with higher scores indicating greater EI. The scale comprises four subscales of four items each, with satisfactory reliability: self-emotions appraisal (SEA, α = 0.92), others' emotions appraisal (OEA, α = 0.86), use of emotion (UOE, α = 0.90), and regulation of emotion (ROE, α = 0.88). Note that the Arabic version has also been validated, showing adequate reliability and validity for use in Arabic-speaking populations (Alkhalaileh et al., 2025).

### Internal locus of control

Internal locus of control, which served as one of the independent variables in the study, was measured based on a seven-item (e.g., "What happens to you in the future mostly depends on you?" "You can do just about anything you really set your mind to," "You often feel helpless in dealing with problems of life" ([reverse-coded]; α = 0.84)) five-point additive scale from Pearlin and Schooler (1978) Sense of Mastery Scale. This scale was designed to measure an individual's general sense of control or mastery over life circumstances and includes both positively and negatively worded statements. Possible answers ranged from 1 to 5. This variable was treated as a continuous variable, with possible total scores ranging from 7 to 35, with higher scores indicating a stronger internal locus of control. To the best of our knowledge, no published Arabic validation exists. Accordingly, we used a translated and back-translated version following the method proposed by Bracken and Barona (1991) to ensure linguistic and conceptual equivalence.

### Control variables

Based on insights from prior research, several control variables were included to account for potential confounding influences and to isolate the unique contribution of our primary predictors to anxiety and emotional exhaustion. These controls were selected to isolate the unique contribution of our primary predictors to anxiety and emotional exhaustion. This method ensured that the observed relationships were not driven by demographic or contextual factors that previous studies have identified as significant determinants of employee well-being. Therefore, controlling for these variables allowed us to more accurately assess the relationships of interest, minimizing the risk of biased estimates. We controlled for demographic variables (age, gender, and marital status) as well as stress-related factors, including life stressors, stress related to the socio-economic crisis, and stress related to the COVID-19 pandemic, all of which have been shown to influence emotional exhaustion levels (Hsu, 2019; Anastasiou and Belios, 2020; Pereira et al., 2021; Wang et al., 2023; Meng and Yang, 2024). Accounting for these factors helped ensure that the observed associations reflected the impact of our key variables, rather than underlying demographic or situational differences among participants. Participants reported their age in years, which was treated as a continuous variable in our analyses. Gender was coded as 0 ("Male") and 1 ("Female"), and marital status was categorized as 0 ("Single") or 1 ("In a relationship"). Both variables were treated as binary in the analysis. Life stressors were assessed using the Holmes-Rahe Stress Inventory, with scores computed as the weighted sum of life change units for all reported events, reflecting the intensity of each stressor (Noone, 2017). Participants were presented with a list of major life events (e.g., "Death of a close family member," "Marital separation") and asked to check all that had occurred within the past year. Each endorsed event contributed to a cumulative stress score, reflecting the overall impact of life changes experienced during that period. This cumulative score was treated as a continuous variable in our analyses. Stress related to the socio-economic crisis was measured using the question: "How has the socio-economic crisis affected your overall stress level?" and was coded as 0 ("The socio-economic crisis decreased my stress level or did not change my stress level") or 1 ("The socio-economic crisis increased my stress level"). Similarly, stress related to the COVID-19 pandemic was measured using the question "How has the COVID-19 crisis affected your overall stress level?" and was coded as 0 ("The COVID-19 crisis decreased my stress level or did not change my stress level") or 1 ("The COVID-19 crisis increased my stress level"). Both variables were treated as binary in our analyses. These context-specific measures were developed by the research team based on our expertise in occupational health and considering the unique circumstances in Lebanon. Although some of these measures were single-item measures, they directly capture participants' perceptions of stress related to these crises, ensuring face validity and contextual relevance. That said, using binary indicators may have limited the range of responses and reduced the sensitivity of these measures to

individual differences in stress intensity. We acknowledge this methodological limitation and have explicitly noted it in the discussion section to reflect the potential loss of information and to encourage future researchers to employ continuous or multi-item scales when feasible.

### Data analysis

Ethical approval for this study was granted by the Institutional Review Boards (IRBs) of the American University of Beirut and the Université du Québec à Montréal. We conducted multiple regression and mediation path analyses using the maximum likelihood (ML) in MPlus version 8.8 (Muthén and Muthén, 2017), following the methodology outlined by Preacher and Hayes (2004). Indirect effects were estimated using 5,000 bootstrap resamples, providing robust confidence intervals (CI) for the mediation effects. All models were adjusted for age, gender, marital status, life stressors, stress related to the socio-economic crisis, and stress related to the COVID-19 pandemic. The primary analytical model examined whether sleep quality (reverse-scored BIS), internal locus of control, and EI indirectly influenced emotional exhaustion through anxiety. This same model allowed us to estimate the direct effects of sleep quality (reverse-scored BIS), internal locus of control, and EI on both anxiety and emotional exhaustion. Although no specific hypotheses were formulated regarding the four individual dimensions of emotional intelligence (i.e., self-emotion appraisal, others' emotion appraisal, use of emotion, and regulation of emotion), an alternative model was also tested, in which the global EI score was replaced by its four components. Although these additional analyses did not provide empirical support for our main hypotheses, they are presented as exploratory findings. Despite no specific hypotheses having been formulated for to their effect, their results offer a more comprehensive understanding of the role of EI– particularly its distinct dimensions – in the studied relationships. A post hoc power analysis was conducted using G*Power 3.1 (Faul et al., 2009) to evaluate the adequacy of our sample size. The analysis indicated that a minimum of 287 participants was required to achieve a statistical power of 0.80 for the main model testing the study hypotheses. As for the exploratory analyses examining each subdimension of EI separately, the required sample size was 311. However, these additional analyses were not subject to hypothesis testing. In line with the general rule of thumb proposed by Tabachnick et al. (2007), the minimum sample size for testing such models should be (50 + 8$n$), where $n$ refers to the number of predictors. As such, the recommended sample size should be 122 for the main model and 146 for the exploratory analyses. Given our final sample size ($n$ = 295), this study was adequately powered.

### Use of artificial intelligence

To improve the grammar and syntax of the manuscript, the authors used OpenAI's ChatGPT (2025 version) as a language editing tool. This assistance was limited to linguistic refinement and did not contribute to the generation of scientific content, data analysis, interpretation of results, or the provision of references or citations. The final manuscript was thoroughly reviewed and validated by the authors to ensure its accuracy, integrity, and to make sure that the intended meaning of all the text was preserved. This process ensured that the manuscript fully reflects the authors' intellectual contributions and scientific judgment.

## Results

### Descriptive analyses

Table 1 presents the descriptive statistics, including means/ proportions and standard deviations. The final sample consisted of 295 participants (61% women, 39% men), with a mean age of 33.19 years (SD = 11.25). Regarding marital status, 46% of respondents were married or living with a partner, while 54% were single. A large proportion of participants (82%) reported experiencing stress related to the ongoing economic crisis, and 70% reported stress linked to the COVID-19 pandemic.

In terms of psychological variables, the mean level of emotional exhaustion was 12.32 (SD = 9.10), and the mean level of anxiety was 9.31 (SD = 6.34). These scores suggest low levels of emotional exhaustion and mild to moderate levels of anxiety (INSPQ, 2025). Average sleep quality (reverse-scored BIS) score was 24.50 (SD = 10.36), while the mean level of life stressors was 148.43 (SD = 102.88). Regarding personal resources, participants reported a mean EI score of 88.60 (SD = 13.60) and a mean internal locus of control score of 23.66 (SD = 5.67).

### Correlational analysis

Table 2 presents the intercorrelations among all study variables. Emotional exhaustion was positively correlated with anxiety ($r$ = 0.46, $p \leq 0.01$) and life stressors ($r$ = 0.25, $p \leq 0.01$). Inversely, emotional exhaustion was negatively correlated with sleep quality (reverse-scored BIS) ($r$ = −0.38, $p \leq 0.01$), EI ($r$ = −0.18, $p \leq 0.01$), and internal locus of control ($r$ = −0.38, $p \leq 0.01$). Additionally, anxiety was negatively correlated with sleep quality (reverse-scored BIS) ($r$ = −0.52, $p \leq 0.01$), EI ($r$ = −0.34, $p \leq 0.01$), and internal locus of control ($r$ = −0.46, $p \leq 0.01$).

EI was positively correlated with its four dimensions: "self-emotion appraisal" ($r$ = 0.77, $p \leq 0.01$), "others-emotion appraisal"

**Table 1.** Descriptive statistics

| | Mean/ Proportion | Standard deviation |
|---|---|---|
| Emotional exhaustion | 12.32 | 9.10 |
| Anxiety | 9.31 | 6.34 |
| Sleep quality | 24.50 | 10.36 |
| Life stressors | 148.43 | 102.88 |
| Emotional intelligence | 88.60 | 13.60 |
| Self-emotion appraisal | 22.28 | 4.62 |
| Others-emotion appraisal | 23.52 | 3.64 |
| Use of emotion | 22.64 | 4.86 |
| Regulation of emotion | 20.16 | 5.27 |
| Internal locus of control | 23.66 | 5.67 |
| Age | 33.19 | 11.25 |
| Gender | 0.61 | - |
| Marital status | 0.46 | - |
| Stress related to the socio-economic crisis | 0.82 | - |
| Stress related to the COVID–19 pandemic | 0.70 | - |

**Table 2.** Correlational statistics

|    | 1 | 2 | 3 | 4 | 5 | 6 | 7 | 8 | 9 | 10 | 11 | 12 | 13 | 14 | 15 |
|----|---|---|---|---|---|---|---|---|---|----|----|----|----|----|----|
| 1  | 1 |   |   |   |   |   |   |   |   |    |    |    |    |    |    |
| 2  | .46** | 1 |   |   |   |   |   |   |   |    |    |    |    |    |    |
| 3  | −.38** | −.52** | 1 |   |   |   |   |   |   |    |    |    |    |    |    |
| 4  | .25** | .25** | −.19** | 1 |   |   |   |   |   |    |    |    |    |    |    |
| 5  | −.18** | −.34** | .17** | .01 | 1 |   |   |   |   |    |    |    |    |    |    |
| 6  | −.16** | −.27** | .16** | .03 | .77** | 1 |   |   |   |    |    |    |    |    |    |
| 7  | .01 | −.09 | .13* | .09 | .62** | .44** | 1 |   |   |    |    |    |    |    |    |
| 8  | −.17** | −.22** | .08 | −.06 | .77** | .44** | .31** | 1 |   |    |    |    |    |    |    |
| 9  | −.18** | −.39** | .14* | −.03 | .76** | .40** | .25** | .48** | 1 |    |    |    |    |    |    |
| 10 | −.38** | −.46** | .33** | −.13* | .54** | .39** | .21** | .55** | .39** | 1 |    |    |    |    |    |
| 11 | −.17** | −.13* | .14* | −.07 | .13* | .04 | .05 | .13* | .14* | .13* | 1 |    |    |    |    |
| 12 | .05 | .06 | .05 | −.06 | −.05 | −.03 | .09 | −.01 | −.15** | −.01 | .06 | 1 |    |    |    |
| 13 | .08 | .03 | −.01 | .04 | .06 | .04 | .10 | .06 | −.02 | .02 | .24** | .03 | 1 |    |    |
| 14 | .22** | .34** | −.20** | .02 | −.25** | −.16** | −.05 | −.24** | −.25** | −.32** | .04 | .09 | .07 | 1 |    |
| 15 | .20** | .28** | −.15** | .09 | −.13* | −.08 | −.02 | −.12* | −.15** | −.19** | −.06 | .15** | −.05 | −.47** | 1 |

*Note:* (a) *p ≤ .05 and **p ≤ .01. (b) 1 = Emotional exhaustion; 2 = Anxiety; 3 = Sleep quality; 4 = Life stressors; 5 = Emotional intelligence; 6 = Self-emotion appraisal; 7 = Others-emotion appraisal; 8 = Use of emotion; 9 = Regulation of emotion; 10 = Internal locus of control; 11 = Age; 12 = Gender; 13 = Marital status; 14 = Stress related to the economic crisis; 15 = Stress related to the COVID-19 pandemic.

($r = 0.62$, $p \leq 0.01$), "use of emotion" ($r = 0.77$, $p \leq 0.01$), and "regulation of emotion" ($r = 0.76$, $p \leq 0.01$). It was also positively correlated with internal locus of control ($r = 0.54$, $p \leq 0.01$). Stresses related to the socio-economic crisis and to the COVID-19 pandemic were positively correlated with emotional exhaustion and anxiety (rs ranging from 0.20 to 0.34, $p \leq 0.01$). Inversely, stresses related to the socio-economic crisis and to the COVID-19 pandemic were negatively correlated with EI and internal locus of control (rs ranging from −0.13 to −0.47, $p \leq 0.05$). All other correlations scores were small and are reported in Table 2.

## Multiple regression analysis

The results regarding the effects of sleep quality (reverse-scored BIS), EI, and internal locus of control on anxiety are presented in Table 3. Consistent with Hypothesis 1, our findings suggest that personal resources (i.e., sleep quality (reverse-scored BIS), EI, and internal locus of control) are directly and negatively associated with anxiety. Specifically, better sleep quality ($b = −0.224$, 95% CI [−0.286, −0.165], $p < 0.01$), higher EI ($b = −0.061$, 95% CI [−0. 112, −0.007], $p < 0.05$), and a stronger internal locus of control ($b = −0.216$, 95% CI [−0. 351, −0.075], $p < 0.01$) were each associated with lower levels of anxiety.

## Mediation analyses

Before reporting indirect effects, we assessed the overall fit of the mediation path analysis model using ML estimation. This model is just-identified (saturated) and includes only observed variables (no latent constructs), meaning that the number of estimated parameters equals the number of observed variances and covariances. As such, fit indices should be interpreted with caution. Nevertheless, the fit indices were Chi-square (11) = 27.47, $p = 0.004$; RMSEA = 0.071 (90% CI: 0.038–0.105); CFI = 0.940; TLI = 0.837; SRMR = 0.058. Indirect effects were estimated using

**Table 3.** Direct effects of personal resources (sleep quality, EI, and internal locus of control) on anxiety and emotional exhaustion controlling for demographics and context-specific stressors (bootstrap 95% CI, unstandardized coefficients)

|  | Anxiety (B [95% CI]) | Emotional exhaustion (b [95% CI]) |
|---|---|---|
| Constant | 21.444 [16.686, 26.166] ** | 16.031 [7.308, 24.283] ** |
| Anxiety |  |  |
| Anxiety |  | 0.327 [0.137, 0.524] ** |
| Sleep quality |  |  |
| Sleep quality | −0.224 [−0.286, −0.165] ** | −0.136 [−0. 234, −0.033] ** |
| Emotional intelligence |  |  |
| Emotional intelligence | −0.061 [−0. 112, −0.007] * | 0.039 [−0.041, 0.112] |
| Internal locus of control |  |  |
| Internal locus of control | −0.216 [−0. 351, −0.075] ** | −0.334 [−0.539, −0.133] ** |

*Note:* *p ≤ 0.05 and **p ≤ 0.01. CI = bootstrap 95% confidence intervals. The following variables were controlled for: age, gender, marital status, stress related to the socio-economic crisis, stress related to the COVID-19 pandemic.

5,000 bootstrap resamples, providing robust CI for the mediation effects. The results pertaining to the indirect effects of sleep quality (reverse-scored BIS), EI, and internal locus of control on emotional exhaustion via anxiety are presented in Table 4. These findings indicate that both sleep quality ($b = −0.073$, 95% CI [−0.125, −0.029], $p = 0.003$) and internal locus of control ($b = −0.071$, 95% CI [−0.140, −0.018], $p = 0.022$) are associated with lower emotional exhaustion through their effects on anxiety.

**Table 4.** Indirect effects of personal resources (sleep quality, emotional intelligence, and internal locus of control) on emotional exhaustion via anxiety controlling for demographics and context-specific stressors (bootstrap 95% CI, unstandardized coefficients)

|  | Indirect effect (b [95% CI]) | p-value |
|---|---|---|
| Sleep quality | | |
| Sleep quality | −0.073 [−0.125, −0.029] | 0.003 |
| Emotional intelligence | | |
| Emotional intelligence | −0.020 [−0.046, 0.002] | 0.078 |
| Internal locus of control | | |
| Internal locus of control | −0.071 [−0.140, −0.018] | 0.022 |

*Note:* CI = bootstrap 95% confidence intervals. The following variables were controlled for: age, gender, marital status, stress related to the socio-economic crisis, stress related to the COVID-19 pandemic.

In contrast, the indirect effect of EI was not statistically significant ($b = -0.020$, 95% CI [−0.046, 0.002], $p = 0.078$). These findings indicate that sleep quality (reverse-scored BIS) and internal locus of control are associated with lower emotional exhaustion via their effects on anxiety, whereas EI does not appear to have a significant indirect effect. These results are partially consistent with Hypothesis 2, suggesting that personal resources are indirectly and negatively associated with emotional exhaustion via their impact on anxiety.

### Supplementary findings

Additional exploratory analyses examining the direct effects of the four distinct dimensions of EI revealed that none of the dimensions had a direct effect on emotional exhaustion. We checked for potential multicollinearity among the four EI dimensions by calculating variance inflation factor (VIF) values. The mean VIF was 1.40, indicating that multicollinearity is not a concern in our analyses. All key variables were also examined for normality, homoscedasticity, and influential cases, supporting the robustness of the analyses. However, the "use of emotion" dimension was associated with higher anxiety levels ($b = 0.152$, 95% CI [0.008, 0.306], $p = 0.044$), whereas the "regulation of emotion" dimension was associated with lower anxiety levels ($b = -0.276$, 95% CI [−0.415, −0.154], $p < 0.001$). Higher anxiety levels were associated with greater emotional exhaustion (see Table 3). Additionally, both sleep quality (reverse-scored BIS) and internal locus of control were associated with lower emotional exhaustion (see Table 3).

Further analyses examining the indirect associations of the four EI dimensions indicated that "regulation of emotion" was associated with lower emotional exhaustion via its association with lower anxiety levels ($b = -0.088$, 95% CI [−0.182, −0.034], $p = 0.013$).

### Discussion

In this study, we sought to examine the relationships among sleep quality (reverse-scored BIS), EI, internal locus of control, and emotional exhaustion, with anxiety considered a variable consistent with a potential mediating role. Interestingly, while two of the three personal resources (i.e., sleep quality and internal locus of control) were directly associated with lower emotional exhaustion, EI did not show this direct association, highlighting the nuanced role of personal resources. This finding is somewhat surprising given prior research (Di Fabio and Kenny, 2016; Guerra-Bustamante et al., 2019;

Moeller et al., 2020; Ordu et al., 2022). It is worth noting that no specific hypotheses were formulated regarding these direct effects. Inversely, the significant associations regarding sleep quality (reverse-scored BIS) and internal locus of control are consistent with prior empirical evidence (Mauss et al., 2013; O'Leary et al., 2017; Parent-Lamarche and Marchand, 2019; Sigurvinsdottir et al., 2020; Shin and Lee, 2021).

As for mediation analyses, they revealed that only sleep quality (reverse-scored BIS) and internal locus of control had significant indirect effects on emotional exhaustion through anxiety. In other words, anxiety showed an indirect association linking these two personal resources to emotional exhaustion, emphasizing their protective role in stressful environments. This was not the case for EI. These findings are consistent with the COR theory (Hobfoll, 1989). More specifically, sleep quality (reverse-scored BIS) and internal locus of control seem to act as protective factors that help prevent the onset of psychological strain. Individuals with strong personal resources are better equipped to regulate their emotional responses to stressors, whereas limited resources appear to increase vulnerability to anxiety. Anxiety may serve as an early indicator of resource depletion that can trigger a downward spiral of escalating strain, ultimately leading to emotional exhaustion. Taken together, these findings confirm Hypothesis 1, which posited that personal resources are directly and negatively associated with anxiety. As for hypothesis 2, which proposed that personal resources are indirectly and negatively associated with emotional exhaustion through their effect on anxiety, it was only partially supported.

Given the unexpected findings related to EI, additional analyses were conducted to examine the effects of its individual dimensions (i.e., self-emotion appraisal, others' emotion appraisal, use of emotion, and regulation of emotion). Previous studies have shown that the different dimensions of EI may have distinct effects on outcomes (e.g., Guerra-Bustamante et al., 2019; Park and Kim, 2021; Parent-Lamarche, 2022), as each dimension captures unique emotional competencies (Mayer et al., 2024). It is therefore possible that the absence of a significant overall effect is due to opposing influences at the dimensional level. Supplementary analyses revealed that certain dimensions of emotional dimensions were significantly associated with tested outcomes and, in some cases, even demonstrated contradictory effects. More specifically, the "use of emotion" dimension was positively associated with anxiety ($b = 0.152$, $p = 0.044$), whereas the "emotion regulation" dimension was negatively associated with anxiety ($b = -0.276$, $p = 0.000$). As for indirect associations, the regulation of emotion dimension was associated with lower emotional exhaustion via its association with anxiety ($b = -0.088$, $p = 0.009$). The "use of emotion" dimension showed no direct effect on tested outcomes. Lastly, the other dimensions of EI (i.e., self-emotion appraisal, others' emotion appraisal) did not have any direct or indirect effects on tested outcomes.

The results obtained from this study are quite interesting and contribute to refining our understanding of EI. It appears that the "regulation of emotion" is the most beneficial skill for maintaining good psychological health, particularly in high-stress contexts such as Lebanon. A recent systematic review has shown that the "regulation of emotion" dimension tends to have stronger effects on health compared to the other facets of EI (Baudry et al., 2018). The ability to regulate one's emotions also seems to be associated with greater happiness (Guerra-Bustamante et al., 2019). This result might be partially explained by the hypersensitivity hypothesis, which suggests that individuals with higher EI experience emotions more intensely (Fiori et al., 2023). It has also been suggested that greater emotional sensitivity is often accompanied by greater

emotional management. Emotionally intelligent individuals not only feel emotions more deeply but are also better equipped to regulate them effectively (Fiori et al., 2023). One recent study found that people scoring high on emotion regulation valued happiness more than anger (Fiori et al., 2024), highlighting their ability to focus on positive emotional experiences. Individuals skilled in emotion regulation tend to experience less anxiety and have fewer ruminative thoughts (Martinez-Pons, 1997). This, in turn, helps support their capacity to maintain positive emotional states (Martínez-Monteagudo et al., 2019) and recover more quickly from distress (Salovey and Mayer, 1990).

Furthermore, and importantly, it seems that certain abilities related to the "use of emotion" dimension may not always be beneficial for reducing anxiety and may lead to emotional exhaustion. This unexpected finding is difficult to explain, as the "use of emotion" refers to the ability to channel emotions constructively to support motivation and performance and is typically considered a positive resource (Salovey and Mayer, 1990). One possible explanation for that effect is that individuals who are highly performance-oriented may experience greater anxiety and, consequently, rely more heavily on this specific emotional competence. Even though Fiori et al. (2024) have not evaluated the specific effect of the "use of emotion" dimension on emotional sensitivity, it is reasonable to assume that individuals scoring high on this ability may be more emotionally sensitive. This heightened sensitivity could increase their susceptibility to stressors and anxiety. Moreover, using emotions to motivate oneself may inadvertently intensify internal pressure, thereby increasing vulnerability to anxiety. Another possible explanation draws on well-established perspectives suggesting that each dimension of EI may have a potential "dark side" (Kilduff et al., 2010). For the "use of emotion" dimension, this dark side may involve the strategic, instrumental use of emotions to achieve important goals. It is well established that authenticity fosters psychological well-being (Ménard and Brunet, 2011; Rivera et al., 2019; Sutton, 2020). When emotions are mobilized in a calculated (i.e., strategic) rather than authentic manner, the resulting internal tension may heighten psychological strain and, in turn, increase anxiety. Another possible explanation could be that individuals experiencing higher anxiety levels might rely more heavily on strategic emotional regulation to manage stressors. From the perspective of self-determination theory, such strategic use of emotion might, in some cases, conflict with intrinsic emotional needs or authentic self-expression. According to the self-determination theory, healthy emotion regulation depends on awareness, volition, and the integrative processing of emotional experiences (Ryan and Deci, 2001; Roth et al., 2019). When emotion regulation becomes overly controlled or externally driven, rather than autonomous, it may undermine authenticity and increase internal tension, in turn, fostering anxiety. In line with this reasoning, Trigueros et al. (2019) showed that controlling environments undermine the satisfaction of basic psychological needs, which in turn hampers emotional integration and authentic self-regulation. Such needs frustration may reduce individuals' ability to regulate their emotions in an adaptive manner, reinforcing internal tension and anxiety. Under such circumstances, the effortful mobilization of emotions could heighten internal tension and anxiety, suggesting a possible "dark side" of EI. Future researchers are advised to explore the conditions under which the use of emotion could shift from being protective to potentially maladaptive.

## Theoretical contribution

The theoretical contribution of this study lies in demonstrating that not all personal resources necessarily lead to psychological well-being or mental health. These findings add important nuances to the COR theory (Hobfoll, 1989). Specifically, one personal resource (i.e., EI), when considered globally, did not have significant effects on our studied outcomes. However, when broken down into its separate dimensions, the results differed. Specifically, EI, when considered globally, did not show significant effects, but its separate dimensions revealed both favorable (regulation of emotion) and detrimental (use of emotion) effects. While some dimensions did not have any significant effects, one had a clear favorable impact (i.e., regulation of emotion), and one dimension (i.e., use of emotion) even showed a detrimental effect on psychological health. Based on these findings, the accumulation of resources is not necessarily always optimal; rather, it is the accumulation of the right resources that proves beneficial. Thus, from a theoretical standpoint, it is crucial to consider the potential "dark sides" of personal resources and acknowledge that not all resources contribute equally to psychological health.

## Practical implications

These study findings clearly demonstrate the favorable effects of sleep quality (reverse-scored BIS) and internal locus of control. For organizations seeking to enhance employee psychological health, interventions targeting these personal resources are recommended. To this end, organizations could offer activities with health professionals providing tips and raising awareness about the importance of sleep quality for optimal functioning. Furthermore, organizing workshops with psychologists to foster the development of an internal locus of control could be highly beneficial. This is particularly true in high-stress contexts such as the one experienced in Lebanon, where socio-economic and political pressures can exacerbate occupational strain. One promising approach is solution-oriented brief counseling. This type of counseling tends to be goal-directed, focused on the future draws on an individual's natural strengths and resilience. By tapping into solutions employees already use in their lives, this approach can help foster their confidence in their own abilities (Musyarofah et al., 2025). The advantage of this resource is that it can be cultivated through interventions (Musyarofah et al., 2025). The goal here would be to help individuals move from an external to an internal locus of control, capitalizing on its well-documented benefits. Moreover, it would also be valuable to focus on developing employees' EI in a similar way, as this is another personal resource that can be developed throughout one's lifespan (Gilar-Corbi et al., 2019). In doing so, it is particularly important to emphasize the emotional regulation aspect of EI while raising awareness of the potential negative effects of instrumental use of emotion. Considering these "dark side" effects, EI training should prioritize emotion regulation and self-awareness over purely goal-directed emotion use. These trainings should foster authentic self-expression and reduce the risk of internally driven tension or anxiety. At the policy level, organizations and occupational health authorities could integrate these findings into broader mental health strategies. Examples of such strategies include embedding resource-building initiatives into employee wellness programs, establishing supportive organizational cultures, and providing access to psychological support services. These measures can not only improve individual well-

being but also foster more resilient and productive workforces in Lebanon's challenging work environment climate.

### Limitations and future research directions

This research presents several limitations that should be acknowledged. First, its cross-sectional nature does not permit us to draw conclusions about the causality between our variables. Although this study explored the direct and indirect roles of sleep quality (reverse-scored BIS), EI, and internal locus of control in shaping anxiety and emotional exhaustion, the temporal direction of these associations remains uncertain. An alternative model could position emotional exhaustion as a mediator and poor sleep quality as the outcome, such that personal resources (e.g., emotional intelligence and internal locus of control) reduce emotional exhaustion, which in turn improves sleep quality. Testing such alternative specifications would provide additional insights into the potentially reciprocal interplay between sleep and psychological health. To clarify how personal resources influence psychological outcomes over time, especially in chronically stressful settings like Lebanon, future research should adopt longitudinal designs. It is also important to note that, because the BIS was reverse-scored for this study, clinical cutoffs for insomnia no longer apply. Future research may benefit from using dedicated sleep quality measures (e.g., Pittsburgh Sleep Quality Index) for more clinically interpretable results.

Another limitation stems from the exclusive use of self-report measures, which may be subject to common method bias. While self-assessments are commonly used in research on psychological health, they remain subjective measures and may be influenced by personal biases, social desirability, or lack of self-awareness. Future research could benefit from incorporating alternative assessment methods, such as evaluations conducted by trained psychologists or other qualified professionals, to strengthen the validity of findings and reduce the risk of perceptual bias inherent in self-reported data. In addition, certain context-specific stressors (i.e., stress related to the socio-economic crisis and to the COVID-19 pandemic) were measured using binary indicators. While this approach reduced response burden in a challenging data collection context, it may have limited the variability captured in participants' experiences. It might have also reduced the sensitivity of these measures to individual differences in perceived stress levels. Future research should consider using continuous or multi-item measures to provide a more nuanced assessment of these context-specific stressors.

The lack of contextual information about participants' workplaces also limits interpretation. Moreover, because participants were recruited using a non-randomized snowball sampling approach, the external validity of the findings is limited, and results should be generalized with caution. This sampling strategy may have introduced referral bias, whereby participants recruit others who share similar characteristics or experiences. As a result, certain industries, occupations, or regions in Lebanon may be over- or under-represented in the sample, which may limit the representativeness of the data and influence the observed associations. We should also note that important work-related variables (e.g., shift/night work, working hours, job demands/control) were not included in our analyses, and their omission may have influenced the observed relationships. Also, we did not examine potential differences based on participants' language (English vs. Arabic) or the format of questionnaire completion (online vs. paper). Future research could implement formal standardization procedures, such as data collector training, order randomization, or attention checks, particularly in bilingual or Arabic-speaking populations, to improve data quality. Researchers are also encouraged to examine the psychometric equivalence and measurement invariance of these instruments across language groups, as well as to collect richer contextual data from diverse organizational settings, to enhance the generalizability and applicability of future findings.

Future studies should also investigate other potential antecedents and conditions that shape the availability and impact of personal resources in the workplace. Importantly, future research should explore the distinct effects of the individual dimensions of EI. Given the nuanced nature of this construct, analyzing its components separately may refine our understanding of how specific emotional abilities contribute to psychological health. This could be particularly useful in identifying which facets are most protective under chronic stress, and in tailoring interventions to strengthen them. Moreover, it is possible that certain facets of emotional intelligence, such as the "use of emotion," could operate differently depending on an individual's emotional state. More specifically, higher anxiety might lead some individuals to rely more on strategic emotional regulation, which could in turn influence psychological outcomes. Future research could explore these potential bidirectional effects. It is also plausible that the positive association observed for the "use of emotion" dimension reflects an overlap with related characteristics such as achievement orientation, perfectionism, trait anxiety, or exposure to demanding job conditions rather than the construct itself. Future studies would benefit from examining these possibilities by testing such factors as moderators or including them as control variables to clarify the unique contribution of this EI facet. Additionally, there is room to explore whether key personal resources, such as EI or locus of control, can be cultivated through targeted interventions. Encouraging resource development may help build resilience against anxiety and emotional exhaustion in high-stress environments. Future research could also explore practical strategies to enhance sleep quality and internal locus of control in Lebanese workplaces, such as workplace wellness programs, stress management workshops, or targeted training initiatives. Promoting such practices could support the creation of healthier, more psychologically sustainable work settings. Moreover, while our model specifications were based on strong theoretical and empirical foundations, future research could consider testing alternative model specifications to further examine the robustness of our findings. This could provide additional insights into how different combinations of independent variables and mediators might influence psychological outcomes, particularly in complex, high-stress contexts. Such approaches may also allow the exploration of potential reverse or context-dependent effects of certain EI dimensions. It is also important to consider the unique Lebanese context in which this study was conducted. Factors such as political instability, socio-economic pressures, and industry-specific challenges may have influenced the observed relationships. While cultural factors (e.g., collectivism, fatalism, religious coping) were not directly assessed in this study, they could be explored in future research to examine how such factors might moderate the effects of personal resources on psychological outcomes. Future research should examine how these contextual factors moderate the effects of personal resources on psychological outcomes, as well as explore the generalizability of our findings to other populations experiencing chronic adversity. Finally, all coefficients reported throughout the manuscript are unstandardized. As a recommendation for future researchers, standardized coefficients could be calculated to allow comparison of effect sizes across variables and provide enhanced practical interpretation. Some of

the supplementary analyses, presented in the "Supplementary Findings" section, were exploratory in nature and were not tied to any specific hypotheses. Please see supplementary findings. These results were included to provide additional insights beyond the primary hypotheses. That said, we would like to reiterate that they should be interpreted with caution and require replication in future studies.

## Conclusions

The primary goal of this study was to examine the relationships between sleep quality (reverse-scored BIS), EI locus of control, and emotional exhaustion, with anxiety as a potential variable consistent with a mediating role. The results emphasize the intricate nature of personal resources and their distinct effects, highlighting that not all resources contribute to positive outcomes in the same way. While sleep quality (reverse-scored BIS) and internal locus of control were found to have beneficial indirect effects on emotional exhaustion through anxiety, EI, especially the "use of emotion" dimension, demonstrated a more complex relationship, with some potentially negative consequences. These findings indicate that the influence of personal resources is not inherently positive, and their effects can vary based on how they are applied. While acknowledging the limitations of the study, our aim is to enrich the ongoing conversation about the role of personal resources in promoting psychological health. The insights gained from this research provide a deeper understanding of how different dimensions of emotional intelligence and other personal resources are connected to anxiety and emotional exhaustion, particularly during times of crisis, such as those experienced in Lebanon. These results underscore the importance of recognizing both the beneficial and potentially harmful aspects of personal resources in interventions designed to enhance individual and organizational psychological health. Considering these findings, it is crucial for organizations and healthcare professionals to focus on cultivating the right personal resources, such as sleep quality and internal locus of control, while also supporting EI development in a balanced way. Specifically, training programs aimed at improving emotional regulation and the constructive use of emotions could be especially helpful. As individuals, organizations, and societies continue to face evolving challenges, whether from global events like the COVID-19 pandemic or the ongoing demands of modern work environments, it is essential to adopt strategies that prioritize the development of appropriate personal resources to sustain psychological health.

**Open peer review.** To view the open peer review materials for this article, please visit http://doi.org/10.1017/gmh.2026.10165.

**Data availability statement.** The study's data are unavailable to the public or inaccessible upon request, as this information could compromise the privacy of the research participants. However, de-identified data, the codebook, and analysis syntax are available from the authors upon reasonable request.

**Acknowledgements.** The authors would like to thank students at the American University of Beirut for their help with the extensive data collection.

**Author contribution.** Annick Parent-Lamarche conceived and designed the study, developed the research protocol, conducted the literature review, analyzed and interpreted the data, and drafted and edited the manuscript.
Sabine Saade conceived and designed the study, developed the research protocol, was responsible for the data collection, and provided critical feedback on the manuscript.

**Financial support.** A lottery system was used whereby participants could earn a small cash prize in exchange for participating in our study. The source of the fund was internal and was provided by the American University of Beirut.

**Competing interests.** The authors declare none.

**Ethics statement.** Ethical approval for this study was obtained from the Institutional Review Boards (IRBs) of the participating organizations (American University of Beirut and Université du Québec à Montréal). All participants provided informed consent prior to participation.

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
