## [Reviewer Report]

Title: The term personal resources is misleading in the title, emotional intelligence and locus of control can be mentioned directly

Page 1: Line 25 - not indirectly - language needs to be modified to reflect if there is a direct association

Page 2: Lines 40 onwards - More synthesis is needed as the focus should be on Lebanon but then switches to Canada. The introduction can be more succinct in it’s ability to capture the focus on Lebanese individuals.

Page 3: Line 8 onwards - Anxiety is the mediating variable and it would make organizational sense to start with the main variables of emotional exhaustion, sleep quality, EI and LOC followed by anxiety.

Page 5-6: There is a mention of adversity in Lebanon without an elaboration, followed by a comment on the US situation. This does not help with focusing the paper to the Lebanese population, thus not making the context clear

H1: based on this, anxiety is a primary variable for the correlation, while not capturing it’s role as a mediating variable. H2 by itself captures the research question of the study

Sample: the statement says that random sample was targeted, not revealing much about how the sample was random; in the next paragraph snow-ball sampling is mentioned - using consistent wording while describing the sampling will help readers understand the methods used better

Control variables - this can be further clarified to make clear that gender, age, marital status were not controlled for, while stress was controlled for? the language can be clearer.

Results - direct statements about findings related to the hypotheses are needed to clearly state how the findings related to the hypotheses

Dimensions of emotional intelligence mentioned in the discussion can be added to the result section under supplementary findings and then further elaborated on in the discussion section

Overall:

Repetitive citations in same paragraphs, indicating the need for further summarizing and synthesizing of the literature to present one’s new idea generated from the read literature

The paper is a good contributor to enhancing personal resources for management of stress and burnout with the mediating effect of anxiety, however, the context of Lebanon and current adversities are not elaborated to understand the results of the study. The diversity of the sample is good, however, providing a sample characteristics table will be helpful to look at a quick summary of the diversity. The organization and flow of the paper can be enhanced, especially for the introduction and literature review to help readers follow the paper effectively.

---

## [Reviewer Report]

Abstract

The purpose stated was not consistent with the main title of the study.

Should mention the nature of the study design, the study location and questionnaires used.

For the result, mention the number of the respondents and the coefficients, 95% CI and the p values.

Introduction

In terms of theoretical grounding, the theoretical rationale connecting emotional exhaustion and the factors that may influenced it is still lacking.

Furthermore, the study’s aims and research questions should be clearly and concisely stated in the introductory section to guide the reader through the study’s purpose and scope.

Methods

The Ethical approval should after the data analysis subheading.

what is the nature of the study design?

Which part of Lebanon?

Were the respondents randomly selected, as in the other statement the investigators using other approaches that included online, snowball approach for which these methods were not randomized.

What was the sample size calculated?

The inclusion and exclusion criteria not clearly stated.

There is lacking in describing related to the point of entry to find the respondents.

What type of these community-based workers? Please define.

More information is needed concerning how key variables were operationalized and validated.

The description so each scale should be clear: how do the scales will be measured? continuous or categorical?

Should mention which were the dependent variable(s) and independent variable(s).

The discussion of the psychometric properties of the scales, including reliability and validity indices, should be expanded to demonstrate methodological rigor.

The choice of analysis must be clearly stated whether to do a variance-based structural equation modeling (SEM) or co-variance based SEM. These two methods are different in their purpose of analysis. Give strong justification on why chose that SEM approach for further analysis.

Results.

The descriptions of respondents’ characteristics should be described first followed by the rest.

Table 1 incorrectly applied.: Please separate the respondents’ socio-demography and the scales findings.

Table 2 and Table 3: both are incorrectly presented to show the SEM findings.

The findings should be described in the text then summarize with tables or graphs whereever applicable.

Discussion

With the correct and reliable method of analyses and findings will provide a good discussion.

In addition, it should be strengthened by providing more nuanced insights into the implications of your findings for policy, organisation planning, or occupational interventions in the broader Lebanese context.

---

## [Reviewer Report]

Thank you for the opportunity to review your manuscript “Exploring the Links Between Workers' Personal Resources, Sleep Quality, and Emotional Exhaustion During Challenging Times: Anxiety as a Mediating Factor.” This study addresses a highly relevant and timely topic, particularly given the challenging socio-economic context in Lebanon. The application of Conservation of Resources theory and the nuanced examination of emotional intelligence dimensions represent valuable contributions to the literature.

Major Methodological Concerns

Sampling and Generalizability Issues

The manuscript contains an important inconsistency regarding sampling methodology. While the abstract states participants were “randomly selected,” the methods section describes snowball sampling combined with online and in-person recruitment. This is not random sampling and should be corrected throughout the manuscript. Please clarify the actual sampling procedure and discuss potential sampling biases and their implications for generalizability. The diverse occupational representation is a strength, but the convenience sampling approach limits external validity claims.

Sample Size and Statistical Power

The manuscript does not clearly report the final sample size, which is crucial for evaluating the adequacy of the statistical analyses. Please provide the exact sample size and consider reporting post-hoc power analyses for the mediation models. Additionally, please report the distribution of participants by language (English/Arabic), completion method (online/paper), and any missing data patterns with their handling procedures.

Measurement Scale Concerns

Several measurement issues require clarification:

Sleep Quality: The Bergen Insomnia Scale typically measures insomnia symptoms (higher scores = worse sleep), not sleep quality as conceptualized in your introduction. Please clarify whether scores were reverse-coded and ensure consistent terminology throughout.

Internal Locus of Control: The description mentions a 7-item, 5-point scale from Rotter (1966), but the original scale uses forced-choice format. Please provide the specific version used, example items, and scoring procedures.

Emotional Intelligence: The Wong & Law scale anchors appear reversed (1=strongly agree, 7=do not agree). Please verify the scoring direction and provide reliability coefficients for each subscale.

Statistical Analysis and Model Specification

The statistical analysis section needs strengthening in several areas:

Report comprehensive model fit indices (CFI, TLI, RMSEA, SRMR) for all structural equation models, not just CFI and TLI

Provide bootstrap confidence intervals for indirect effects rather than relying solely on p-values

Given the ordinal nature of many variables, consider robustness checks using WLSMV estimator

Address potential multicollinearity among emotional intelligence dimensions by reporting VIF values

Consider testing alternative model specifications to ensure robustness of findings

Theoretical and Interpretive Issues

Cross-Sectional Design Limitations

While acknowledged as a limitation, the cross-sectional design significantly constrains causal interpretations of the mediation model. The bidirectional relationship between sleep and psychological health, which you mention in the introduction, particularly challenges the assumed directionality. Consider discussing alternative theoretical models or providing stronger caveats about causal inference limitations.

Emotional Intelligence Interpretation

The finding that “use of emotion” was associated with higher anxiety levels is intriguing but requires deeper theoretical exploration. While you propose several explanations (performance orientation, strategic use), consider also discussing the possibility of reverse causation—that individuals experiencing higher anxiety may rely more heavily on strategic emotional regulation. The “dark side” interpretation would benefit from stronger theoretical grounding, perhaps drawing from authenticity or self-determination theory.

Technical Recommendations

Translation and Cultural Adaptation

While the translation procedure is described, please clarify whether these scales have been previously validated in Arabic-speaking populations. If not, consider reporting measurement invariance analyses across language groups to ensure equivalence. At minimum, report reliability coefficients and descriptive statistics by language.

Control Variables and Confounding

The selection of control variables is reasonable, but consider expanding the discussion of their rationale and effects. Some variables particularly relevant to the Lebanese context (COVID-19 stress, socio-economic crisis stress) are measured as binary indicators, which may lose important information. Consider using continuous measures if available.

Multiple Comparisons

The exploratory analyses of emotional intelligence dimensions involve multiple comparisons without adjustment. Consider applying false discovery rate correction or clearly labeling these as exploratory findings requiring replication.

Practical Implications and Future Directions

Intervention Specificity

While the practical recommendations are sound, they could be more specific. For example, what types of workshops would most effectively develop internal locus of control? Given the potentially negative effects of strategic emotion use, how should emotional intelligence training be structured to emphasize regulation while minimizing instrumental use?

Contextual Considerations

The unique Lebanese context is central to this study’s significance. Future research could explore how specific contextual factors (political instability, industry-specific pressures) moderate these relationships. Consider discussing how findings might apply to other populations experiencing chronic adversity.

Minor Editorial Issues

Tables and figures are referenced but not included in the manuscript

Some citations appear to be from non-peer-reviewed sources (Statista, news reports) that could be replaced with academic references

The AI use disclosure is appropriate but could include more details about verification procedures

Overall Assessment

This study makes an important contribution to understanding personal resources in high-stress contexts, particularly through its nuanced examination of emotional intelligence dimensions. The theoretical refinement of COR theory—showing that not all personal resources have uniformly positive effects—is valuable. However, the methodological limitations, particularly regarding sampling, measurement clarity, and statistical reporting, need to be addressed before publication.

The findings regarding the differential effects of emotional intelligence dimensions are particularly noteworthy and challenge conventional assumptions about this construct. With appropriate revisions addressing the methodological concerns, this manuscript would make a solid contribution to the literature on occupational health and resilience.

Recommendation: Major Revisions Required

The study addresses an important topic with valuable findings, but significant methodological clarifications and improvements are needed before it can be considered for publication.

---

## [Reviewer Report]

Overall assessment: This is an important and timely study examining sleep quality, internal locus of control, and emotional intelligence (EI) in relation to anxiety and emotional exhaustion in a high-stress Lebanese context. The analyses are generally appropriate, and the exploratory dive into EI subdimensions is interesting. However, there are several issues that require correction before publication, including sampling description inconsistencies, measurement clarity, reporting consistency, and interpretation of model fit. I recommend major revisions.

#1 Study design and sampling inconsistency; generalizability

- The abstract on p.1 states the sample was “randomly selected,” whereas the Methods clearly describe a non-probabilistic snowball sampling approach. Please align these descriptions (use snowball sampling consistently).

- Expand the limitations to explicitly address representativeness concerns (referral bias, potential industry/region skew).

#2 Anxiety scale reference mismatch

- You report using the GAD-7 (7 items), but cite an Arabic validation for the “GAD-5” (Sawma 2025). Either cite an Arabic validation for GAD-7 or clarify if GAD-5 was used. As written, there is a validity gap.

#3 Sleep scale (Bergen Insomnia Scale) item count, range, and reverse scoring

- The Bergen Insomnia Scale (BIS) is typically 6 items scored 0–7 days (total 0–42). You report “7 items” and a total range of 0–49. Please specify the exact version used or the rationale for an added item.

- Detail the reverse-scoring procedure at the item level and report reliability (alpha) after reversal. Because clinical cutoffs for BIS won’t apply after reversal, use a consistent label such as “sleep quality (reverse-scored BIS)” and consider noting in limitations that a dedicated sleep quality measure (e.g., PSQI) might be preferable.

#4 Internal locus of control scale source and format

- Rotter (1966) is a 29-item forced-choice instrument. You used a 7-item, 5-point Likert short form. Please provide the specific short-form source/citation and describe item selection and validation rationale.

#5 Emotional exhaustion instrument and Arabic validation

- You used the MBI-GS emotional exhaustion subscale (5 items), but cite Arabic validation for MBI-HSS (health services). These are not identical. Please add an Arabic validation reference for MBI-GS, or clearly state that you used your translated/back-translated version and report its psychometrics.

#6 Model fit indices and possible saturation

- Reporting CFI/TLI=1.000, RMSEA=0.000, SRMR=0.000 suggests a just-identified (saturated) model where fit indices are uninformative. Please note this explicitly and avoid overinterpreting fit. Also, report the bootstrap resample count (e.g., 5,000) used for indirect effects.

#7 Inconsistencies between tables and text (must fix)

- Table 3 shows EI → emotional exhaustion as 0.039 [0.012, 0.136], which appears significant (CI excludes 0), yet there’s no significance marker and the text states “no direct effect.” One of these is incorrect—please reconcile.

- Supplementary analysis: “Use of Emotion → anxiety” is reported with CI [0.008, 0.306] but p=0.44 in one place and p=0.039 in another. Likely a typo (“0.044”). Please correct throughout.

- Unify reporting of standardized (β) versus unstandardized (B) coefficients. Clearly label in tables and text and use consistent notation.

#8 Missing data handling

- You state listwise deletion, but using Mplus with MLR allows FIML, which is typically preferable and less biased under MAR. Consider re-estimating with FIML or justify listwise deletion after testing missingness (e.g., Little’s MCAR test). Alternatively, consider multiple imputation.

#9 Confounding and workplace context

- Covariate adjustment is sensible (age, gender, marital status, stressors), but important work-related confounders are not included (e.g., shift/night work, working hours, job demands/control, industry/occupation fixed effects). If available, add sensitivity analyses; otherwise, strengthen the limitation about omitted confounding.

#10 Causal interpretation of mediation in cross-sectional data

- Expressions such as “anxiety mediates” risk causal overreach in a cross-sectional design. Please temper language to “consistent with mediation” or “indirect association,” and emphasize temporal ambiguity. Your mention of alternative specifications is good—consider adding a brief sensitivity test or more concrete plans for longitudinal/intervention research.

— Desirable but not essential enhancements —

#11 Effect sizes and practical significance

- Provide standardized effects and, where possible, translate effects into practical terms (e.g., a 1 SD increase in sleep quality is associated with X decrease in anxiety and Y decrease in emotional exhaustion) to inform practitioners.

#12 Holmes–Rahe stress inventory scoring

- Clarify whether you used the weighted Life Change Units (LCU) sum or a simple count. The reported mean (148.43) suggests LCU; make this explicit and cite the scoring method.

#13 Consistency of titles/terminology and language polish

- Ensure consistent use of “personal resources” vs. listing specific constructs in the title/abstract. Address minor typos (e.g., possessives like individuals’). Your AI-use disclosure is appropriate; ensure it complies with the journal’s policy and place it where required.

#14 Data sharing and reproducibility

- Even if raw data cannot be shared, consider sharing de-identified aggregates, codebooks, and analysis code (e.g., Mplus syntax) in a repository, or note that such materials are available on reasonable request to improve reproducibility.

#15 Theoretical nuance on the “dark side” of Use of Emotion

- The interpretation is thought-provoking. Also acknowledge measurement/construct confounding possibilities (e.g., achievement orientation, perfectionism, trait anxiety, job demands) that could drive the positive association with anxiety. Propose these as moderators/controls for future work.

If these issues are addressed—particularly the measurement and reporting inconsistencies—the manuscript’s contribution would be substantially strengthened and suitable for publication.

---

## [Reviewer Report]

Thank you for addressing all the comments succinctly and effectively. The paper reads very well at this point. Some additional minor observations are made below:

In the theoretical basis, anxiety is mentioned as the DV while with methods, it is mentioned as a mediator, Clarity should be maintained throughout given you have 2 hypotheses treating anxiety differently. Similarly, consistency of terms is needed, emotional exhaustion is mentioned in the discussion (linking it to anxiety). Given there are multiple variables, please ensure same terms are used to refer to the variables throughout the paper.

---

## [Reviewer Report]

Abstract

Should added the coefficient and 95% CI values in the result.

Introduction

Though emphasizes national crises in Lebanon (e.g., 77.78% burnout, 25.8% anxiety prevalence), yet lacks specifics on occupational stressors, job conditions, or why community-based workers are vulnerable.

“Community-based workers” is undefined, with no details on sectors, roles, or unique vulnerabilities.

While the introduction establishes that anxiety may mediate relationships between personal resources and emotional exhaustion, it provides limited theoretical justification for why anxiety specifically should be the mediator rather than other psychological states (e.g., stress, depression).

Methods

Bilingual (English/Arabic) administration via paper and online modes is noted, but no protocols for standardization (e.g., data collector training, order randomization) or validity checks (e.g., attention probes) are described.

As for the measurement tools such as Bergen Insomnia Scale, need further confirmatory analysis.

The listwise deletion for missing data (8-11% missingness) was employed without testing assumptions (e.g., MCAR) or alternatives like multiple imputation.

Results

Several critical gaps and missing information that undermine its methodological rigor, interpretability, and applicability:

(1) descriptive reporting: No breakdown of demographic characteristics by sector or occupation despite 19 job sectors, limiting sample heterogeneity understanding;

(2) statistical validation: The results omit crucial validation of statistical assumptions;

(3) effect sizes are inadequately reported and interpreted, limiting practical understanding;

(4) title of the table is not adequately stated.

Discussion

The discussion mentioned “targeting sleep quality” and “fostering internal locus of control” but provides no specific guidance on how to implement these in Lebanese workplaces.

Despite the unique Lebanese crisis context, the discussion makes minimal reference to cultural factors (e.g., collectivism, fatalism, religious coping) that could explain why certain personal resources work differently here than in Western samples.

The mediation model reports perfect fit indices (CFI=1.000, TLI=1.000, RMSEA=0.000, SRMR=0.000), which is statistically impossible and indicates overfitting or model misspecification. The discussion completely ignores this.

---

## [Editor Report]

All authors confirm that this is an important topic. Please attend to further comments by reviewers. In particular pay attention to the methodological and statistical analysis errors identified by reviewer one. These need to be addressed before the manuscript is viable for publication.

---

## [Reviewer Report]

The authors have responded appropriately and accurately to the reviewers’ comments; therefore, I judge that the manuscript is suitable for acceptance.

---

## [Reviewer Report]

The revisions have been addressed effectively and the manuscript reads well now. Thank you for your valuable time on researching this topic.